# Is Less Always More? A Prospective Two-Centre Study Addressing Clinical Outcomes in Leadless versus Transvenous Single-Chamber Pacemaker Recipients

**DOI:** 10.3390/jcm11206071

**Published:** 2022-10-14

**Authors:** Michele Bertelli, Sebastiano Toniolo, Matteo Ziacchi, Alessio Gasperetti, Marco Schiavone, Roberto Arosio, Claudio Capobianco, Gianfranco Mitacchione, Giovanni Statuto, Andrea Angeletti, Cristian Martignani, Igor Diemberger, Giovanni Battista Forleo, Mauro Biffi

**Affiliations:** 1IRCCS Azienda Ospedaliero-Universitaria di Bologna, 40122 Bologna, Italy; 2Unità Operativa di Cardiologia, ASST-Fatebenefratelli-Sacco, Ospedale Luigi Sacco University, 20157 Milano, Italy

**Keywords:** VVIR pacemaker, leadless pacemaker, patients’ selection, complications, clinical outcome

## Abstract

(1) Background: Leadless (LL) stimulation is perceived to lower surgical, vascular, and lead-related complications compared to transvenous (TV) pacemakers, yet controlled studies are lacking and real-life experience is non-conclusive. (2) Aim: To prospectively analyse survival and complication rates in leadless versus transvenous VVIR pacemakers. (3) Methods: Prospective analysis of mortality and complications in 344 consecutive VVIR TV and LL pacemaker recipients between June 2015 and May 2021. Indications for VVIR pacing were “slow” AF, atrio-ventricular block in AF or in sinus rhythm in bedridden cognitively impaired patients. LL indication was based on individualised clinical judgement. (4) Results: 72 patients received LL and 272 TV VVIR pacemakers. LL pacemaker indications included ongoing/expected chronic haemodialysis, superior venous access issues, active lifestyle with low pacing percentage expected, frailty causing high bleeding/infectious risk, previous valvular endocarditis, or device infection requiring extraction. No significant difference in the overall acute and long-term complication rate was observed between LL and TV cohorts, with greater mortality occurring in TV due to selection of older patients. (5) Conclusions: Given the low complication rate and life expectancy in this contemporary VVIR cohort, extending LL indications to all VVIR candidates is unlikely to provide clear-cut benefits. Considering the higher costs of LL technology, careful patient selection is mandatory for LL PMs to become advantageous, i.e., in the presence of vascular access issues, high bleeding/infectious risk, and long life expectancy, rendering lead-related issues and repeated surgery relevant in the long-term perspective.

## 1. Introduction

Single-chamber VVIR pacing constitutes the mainstay treatment of atrio-ventricular block in the presence of atrial fibrillation (AF), representing 10–15% of all pacemaker (PM) implants in Western countries [1]. VVIR candidates are typically older and with more comorbidities, thus having a lower life-expectancy than their dual-chamber counterparts [2]. Implanting an intravascular lead and creating a subcutaneous pocket is still burdened by a number of potential short- and long-term complications, namely, bleeding, infection, and pneumothorax, which increase in incidence and morbidity with patient age [3]. The introduction of leadless (LL) PMs, delivered via transfemoral venous route, appeared to address these issues by providing right-ventricular (RV) stimulation without the need for an intravascular lead or a subcutaneous pocket [4]. While safety and efficacy profiles of LL technology have been characterised in previous studies [5,6,7], it is still unclear whether its application to all VVIR candidates does provide a substantial complication- and survival-related advantage compared to its transvenous (TV) counterpart. To date, no randomised trial comparing LL and TV VVIR outcomes has been conducted, with clinical evidence being limited to prospective or retrospective studies. Furthermore, earlier studies have based their comparison on historical TV cohorts associated with higher complication rates than contemporary ones, thereby likely overestimating any benefit from LL PMs [5,6]. Indeed, the TV PM safety profile is continuously improving, owing to awareness of best practice recommendations and standardisation of training [8,9,10]. This is reflected by more recent LL studies comparing outcomes with contemporary TV VVIR cohorts, which overall failed to demonstrate a clearcut advantage in LL patients [7,11], and applies also to the large comparative analysis of TV and LL outcomes obtained from the Medicare database over a 2-year follow-up period (6219 LL vs. 10,212 TV patients) [12,13]. While demonstrating lower rates of reintervention and overall complications (albeit higher rates of pericardial effusion) in LL PMs, the study failed to demonstrate any difference in 30-day and 2-year all-cause mortality [12]. As a result, current evidence points to a rather mixed picture concerning complication and survival benefits in LL compared to traditional TV VVIR PMs. Owing to the limited life expectancy of VVIR recipients and to the time-dependent increase of lead-related issues [3,14], it can be argued that the majority of VVIR candidates are unlikely to have sufficient life expectancy to be negatively impacted by lead- or pocket-related risks. Owing to the significantly higher costs of leadless technology (10 times a TV system in our country), a real-world prospective assessment of the true scope of VVIR LL application in current clinical practice is needed. For these reasons, we aimed to observe whether utilisation of TV and LL based on clinical judgement would prove clinically effective in minimising overall complications in VVIR PM recipients.

## 2. Materials and Methods

This is a prospective study of 344 consecutive patients undergoing either LL (Micra™, Medtronic Inc., Minneapolis, MN, USA) or TV VVIR pacemaker implantation in two tertiary cardiology centres in Italy between June 2015 (Micra™ market release in Italy) and May 2021. We planned this observational study at the beginning of our experience with LL systems on all consecutive VVIR recipients at our centres. Indications for VVIR pacing included “slow conducted” AF, atrio-ventricular block with comorbid AF (either permanent or accepted as “destination rhythm”) or, in a minority of cases, with sinus rhythm in bedridden cognitively impaired patients. Assignment to either LL or TV group was based on clinical judgement. In particular, LL was favoured in the presence of ongoing or expected chronic haemodialysis, superior venous access issues such as occlusion or need for its preservation, active lifestyle with low amount of pacing expected, frailty causing high bleeding and infectious risk (defined as at least 2 amongst: combined anticoagulation and antiplatelet therapy that could not be interrupted due to high thromboembolic risk, ongoing long-term steroid therapy, ongoing chemotherapy, BMI < 18.5 kg/m^2^), valvular endocarditis or implantable electronic device infection treated by lead extraction within the previous 6 months.

Lead insertion in TV was performed either via cephalic vein cutdown or subclavian/axillary vein puncture. The latter occurred at operator’s discretion, blinded or under fluoroscopic guidance. TV leads were placed either in the RV apex or septum at the operator’s preference and position was confirmed by right/left anterior oblique, or latero-lateral radiographic views as per EHRA expert consensus guidelines (8). LL devices, in turn, were deployed via transfemoral vein route following the manufacturer’s recommendations with vein closure achieved by purse-string suture followed by 5-min manual compression.

Anticoagulation therapy was not interrupted in any patient unless dictated by specific clinical indications. Specifically, vitamin K antagonists were tapered to the INR range of 2–2.5 for the week following device implantation in patients with CHA2DS2-VASc score 3. Direct oral anticoagulants (DOAC) were held on the day of implantation and resumed 24 h thereafter. Dual antiplatelet therapy was not interrupted. Suction drains were used in TV recipients to prevent pocket haematoma in case of dual antiplatelet, anticoagulant plus antiplatelet therapy, incomplete haemostasis, or unstable/high (>2.5) INR, and removed at bleeding cessation on desired antithrombotic regimen (usually within 48–72 h).

As per local protocol, antibiotic prophylaxis consisted of 2 g cefazolin or 1200 mg clindamycin, in penicillin allergic patients, 30 min prior to skin incision. In patients with hospitalisation duration greater than 7 days, 1–2 g vancomycin (dose adjusted to renal function) was used 90–120 min prior to skin incision. In patients with suction drain, in turn, prophylaxis was continued until drain removal. None of the TV patients received a TYRX™ antibiotic envelope.

Device and wound status were assessed 2–3 weeks after hospital discharge. Subsequently, clinical follow-up was performed at 6 months and twice yearly thereafter, compiling data on patient status and pacemaker performance. Stimulation percentage was weighed against time elapsed between device interrogations. Clinical information, including incurring medical events related to comorbidities and death, was obtained during subsequent hospital admissions, ambulatory visits, remote patient follow-up, or by consulting regional telematic databases (in patients unable to attend ambulatory visits).

Data from the TV and LL cohorts were analysed with IBM SPSS Statistics software (26.0 version for Windows; IBM Corp; Armonk, NY, USA). Variable distribution was assessed using the Shapiro–Wilks test. Continuous variables were then evaluated with the Mann–Whitney U test for non-parametric data, while the Kruskal–Wallis test was used when comparing more than two groups. Categorical variables, in turn, were compared with Chi-square tests of independence between groups. Thereafter, survival analyses for both all-cause and cardiovascular mortality were conducted using Kaplan–Meier plots and the log-rank test. Multivariate analyses of all-cause mortality were performed using Cox regression. The level of statistical significance for all the above was set at *p* < 0.05.

## 3. Results

### 3.1. Baseline Characteristics

Seventy-two patients (20.9%) received LL and 272 (79.1%) TV VVIR pacemakers (Table 1). Patients in the LL group were significantly younger than in the TV group (median age 79.5 ± 2.5 vs. 85.0 ± 1.0), with diabetes and ongoing haemodialysis being more prevalent (26.4 vs. 18.0% and 6.9 vs. 0.7%, respectively), while chronic kidney disease (GFR < 60 mL/min/1.73 m^2^) was more common in the TV group (57.7 vs. 36.1%), as reported in Table 1. Atrial fibrillation, either permanent or paroxysmal, was significantly more frequent in the TV group (96.3 vs. 80.6%). No other significant difference in comorbidities (i.e., hypertension, ischaemic heart disease, left ventricular ejection fraction, history of percutaneous or surgical treatment of valvular heart disease) was observed between groups (Table 1). Amongst the 72 LL recipients, five (6.9%) were on haemodialysis, eight (11.1%) had a completely occluded subclavian vein while seven (9.7%) needed superior vein patency maintenance, 16 (22.2%) were active patients with only sporadic pacing required to prevent long pauses, 17 (23.6%) were frail based on the aforementioned criteria, two (2.8%) had recent surgery due to valvular endocarditis, and four (5.6%) had undergone CIED extraction because of pocket or endovascular CIED infection.

### 3.2. Procedural Data

Both procedural and fluoroscopy times were significantly longer in LL compared to TV patients (median procedural time 74 ± 13 vs. 55 ± 5 min, *p* < 0.01; median fluoroscopy time 8.4 ± 2.3 vs. 3.0 ± 1.0 min, *p* < 0.01). In the LL group, most implants occurred in the interventricular septum (88.9%) and only a minority (11.1%) in the right ventricular apex. In the TV group, in turn, the majority of implants were either in the right ventricular apex (50.7%) or interventricular septum (42.7%), with only some in the His bundle (4.8%), right ventricular outflow tract (0.4%), or left ventricle via a coronary vein (1.1%). Fluoroscopy times were not different for septal or apical implants in either group (Figure 1). In terms of electrical parameters at implantation, no statistically significant difference was observed between LL and TV groups in the mean capture threshold (0.75 ± 0.08 V vs. 0.69 ± 0.04 V), impedance (748 ± 28 vs. 698 ± 15 Ohm), and mean intrinsic R wave amplitude (9.8 ± 0.6 vs. 10.8 ± 0.4 mV) (Table 2).

### 3.3. Data at Follow-Up

Mean follow-up times for LL and TV groups were comparable (22.8 ± 2.6 months vs. 23.7 ± 1.1 months, *p* = 0.31). Overall, no LL patients required pacing system revision while six patients (2.2%) in the TV groups required either lead repositioning (five cases) or addition (one case) because of a high pacing threshold. Both groups demonstrated stability of electrical parameters, with only nine patients in the TV group (3.4%) and two in the LL group (2.7%) having an increase greater than 1 V in capture threshold (Table 2).

### 3.4. Complication Analysis

On analysis of periprocedural and long-term complications, no significant difference in the individual and overall complication rate over the entire follow-up period was observed between LL and TV groups (overall acute complications: 5.6 vs. 5.1%, *p* = 0.33; overall long-term complications: 0 vs. 1.9%, *p* = 0.25). The three groin haematomas occurring in the LL group involved one patient on DOAC (apixaban) and two patients on warfarin (pre-procedural INR 2.3 and 1.44, respectively). Regarding haematomas occurring in the TV population, one occurred in a patient on rivaroxaban and aspirin, and in two cases in patients on warfarin with pre-procedural INR 2.29 and 2, respectively, the latter being the only case requiring surgical revision. Overall, six pneumothoraces (2.2%) occurred in the TV group, when performing either blinded puncture of the subclavian vein (three cases) or under fluoroscopic guidance in the postero-anterior (PA) view (three cases). Of the latter, one occurred after crossing to the contralateral side because of an occluded subclavian vein. Only one out of six pneumothoraces required drainage. No pneumothorax occurred in the 68 patients whose vein access was obtained via the cephalic vein or by axillary puncture under 35°-caudal-tilt fluoroscopic guidance in PA view. Four TV PM recipients had mild pericardial effusion at hospital discharge, lead placement being in all these cases in the right ventricular apex; in two of these, no echocardiogram was available in the 2 weeks before implant, thus raising uncertainty on the true correlation with the procedure. The single case of tamponade occurred because of inadvertent RV free wall placement while aiming at the interventricular septum; no long-term sequelae occurred to this patient. Lastly, the single episode of dislocation in the LL group occurred in a patient during device placement in the right ventricular apex, as the 18th consecutive case of the most experienced operator: multiple deployments (>6) were attempted because of thrombus formation at the cup of the delivery catheter preventing ventricular capture at threshold measurements. When capture was eventually obtained, anchoring was stable on two tines only. As a result, the Micra unit dislodged and was retrieved by means of a 2-snares technique to turn it straight as to enter the large-bore sheath (Appendix A online). At follow-up, no LL patient, but six patients in the TV group, underwent lead repositioning (five cases) or lead addition (one case) because of a pacing threshold increase above 2V@04ms to maximise pacemaker longevity.

### 3.5. Mortality and Multivariate Analysis

By Kaplan–Meyer survival analysis, TV recipients had greater all-cause and cardiovascular mortality (Figure 2). Patients from both the LL and TV group were stratified based on stimulation percentage as assessed during device interrogation at last follow-up. Non-cardiovascular as well as cardiovascular mortality were then plotted across five quintiles of stimulation percentage in both groups (Figure 3). On multivariate analysis of mortality in TV and LL cohorts, only age in the TV group and ventricular stimulation percentage in the LL group appeared to significantly impact mortality, while on analysis of the entire population, only age displayed a significant association with mortality (Table 3).

## 4. Discussion

From this dual centre prospective study, a number of conclusions can be drawn on the current clinical application of leadless versus transvenous pacemakers in the setting of VVIR pacing. Given the lack of evidence from randomised trials on LL and TV outcomes, pacing system selection in this study was based on the principle of individualised therapy and resulted in similar complication rates in the two arms. Another important observation is that mortality of VVIR recipients is such that exposure to lead-related unwanted events in TV recipients is negligible, thus likely blunting any expected advantage of LL technology. Factors such as lack of or necessity to spare thoracic veins for future intravenous therapy, high infectious or surgical risks (due to bleeding or lead-related complications) related to clinical frailty, and long-term exposure to implanted pacing system in patients with relatively long life expectancy and active lifestyle, were all considered to favour LL pacing. According to this pre-defined strategy, LL recipients were either those with a more challenging clinical profile or minimal probability of long-term pacing dependence. As a result, they were on average younger than their TV counterparts, which contributed to a lower mortality. A particularly relevant group were patients on permanent haemodialysis where a transvenous approach is often precluded by either superior vein occupation by indwelling catheters or need for their preservation for dialysis treatment, a setting where LL VVIR safety has been confirmed by a recent retrospective study [15].

In agreement with the current literature, our study confirms the low complication rate in TV VVIR recipients, with a substantial reduction relative to historical transvenous cohorts and with no significant difference compared to leadless recipients both in terms of perioperative and long-term complications. Indeed, a recent meta-analysis on four studies including TV control cohorts (344 LL vs. 400 TV patients in total) showed no difference in incidence of haematoma, pericardial effusion, device dislocation, any complication, and death between LL and TV PMs, with a 3.11% pooled complication rate [16]. Importantly, the latter figure, comparable to our study, corroborates the notion that contemporary TV complication rates are significantly lower than historical ones [6,17]. Similar findings also apply to another recent meta-analysis where lower complication rates in the LL group were observed only when compared with historical TV cohorts, while analysis of studies with contemporary TV cohorts revealed a higher 1-year complication rate in LL VVIR recipients [18].

The improvement in TV outcomes over recent years is likely due to increasing efforts aimed at standardising implantation techniques as well as training programmes [8]. Indeed, one of the main challenges of comparing outcomes in LL and TV VVIR PMs lies in the different learning pathways of these two procedures. On the one hand, MicraTM release involved specific training of a limited number of very experienced implanters who would later provide guidance to further operators, with a consequent predictable learning curve, leading to declining complication rates between Investigational Device Exemption (IDE) and later real-world studies [5,6]. On the other hand, transvenous lead implantation has evolved over decades, with diversity in implantation techniques across implanters and thus greater challenges in procedural standardisation. Clear examples of the potential benefits of TV technique standardisation provided in our study are the absence of pneumothorax when adopting a 35° caudal tilt axillary puncture and, on the contrary, increasing risk of inadvertent RV perforation and tamponade when omitting radiologic confirmation of true septal lead placement. Only recently, practical recommendations on CIED surgery, vascular access, and lead placement have been published with the aim of maximising lead performance and minimising both short and long-term complications [8]. Given the limited standardising efforts conducted so far, these recommendations are expected to significantly impact clinicians’ proficiency in the next few years. Examples of the potential benefits of such efforts, detailed in the EHRA expert consensus document, are avoidance of pneumothorax by widespread cephalic vein use or ultrasound-guided vein puncture (not used in this study), or minimisation of inadvertent RV free wall perforation risk by use of specific fluoroscopic views [8]. These simple behavioural changes have been shown to bring down TV complications to 1.5%, possibly impacting long-term outcomes as well. In further support of the dramatic impact that procedural standardisation can bring about, Ahsan et al. demonstrated that the application of a simple infection-control protocol was able to halve CIED infections [10]. 

An additional point of reflection is hinted at by the mortality analysis of our VVIR cohorts, confirming an extremely low life expectancy of VVIR candidates, mostly driven by non-cardiovascular causes. In keeping with our patient selection criteria, patients in the LL cohort were younger and had thus a significantly longer life expectancy on average. In the setting of such limited life expectancy, long-term lead-related and infective risks, which are known to be time-dependent [14], are unlikely to be fully expressed and thus to have significant prognostic relevance, thereby undermining the most significant advantage of LL systems. From the data collected at follow-up, one of the advantages of LL devices appeared to be the lower, albeit not statistically significant, rate of reintervention. This observation, in line with a recent study by El-Chami et al. [12], highlights the importance of selecting patients with sufficient life expectancy to significantly benefit from a lower long-term reintervention rate in LL PMs. Nonetheless, lead-related reintervention needs to be considered in the correct perspective since these were performed to correct for high pacing thresholds, with a view to maximise device longevity. While the unanticipated increase of pacing threshold is well known in literature and lead manipulation is feasible in TV devices [19], it is not in LL pacers, and demands a new device implantation with far superior costs, suggesting a significant bias towards reintervention in the TV compared to LL PMs. On further analysis of survival data, we found that a higher amount of stimulation identifies patients with limited life expectancy where long-term lead-related risks are likely negligible. This finding further corroborates our selection criterion whereby patients with expected pacing-dependence underwent TV implantation. Our study also confirms the excellent long-term electrical performance of LL devices. This information, together with recent work putting forward a number of predictors of poor LL electrical performance, suggest that LL long-term performance is not only highly efficient but has also a significant degree of predictability [20]. Lastly, our multivariate analysis on survival did not identify any significant factor associated with all-cause mortality at follow-up, with the exception of age, whose associated comorbidities pose a heavy burden of non-cardiovascular mortality (Table 3, Figure 2 and Figure 3). 

In summary, the main advantages of LL devices appear to lie in the decreased long-term reintervention rates, whereas benefits in terms of short-term complications appear far less clear-cut [6,9,10,11,12]. Our experience confirms the usefulness of our clinically-oriented implantation strategy when aiming for “individualised” therapy with a view to minimise both short- and long-term complications. Moreover, in the long-term perspective, the shortcomings of LL pacemakers, such as lack of physiologic ventricular activation as enabled by conduction system pacing, limited diagnostics, and absence of wireless remote patient monitoring, may be detrimental for patients’ management [13].

## 5. Conclusions

Given the limited complication rates observed in this contemporary single-chamber TV cohort and low life expectancy of this population, extending LL indications to all VVIR recipients is unlikely to provide a clearcut clinical benefit to this particular patient group. Considering the higher costs of LL technology, these data prompt a careful selection of patients most likely to gain a real advantage from LL pacemakers. In addition to the setting of vascular access issues and high bleeding or infectious risk, these may include patients with sufficient life expectancy for lead issues and repeated surgery to become relevant in the long-term perspective.

## Figures and Tables

**Figure 1 jcm-11-06071-f001:**
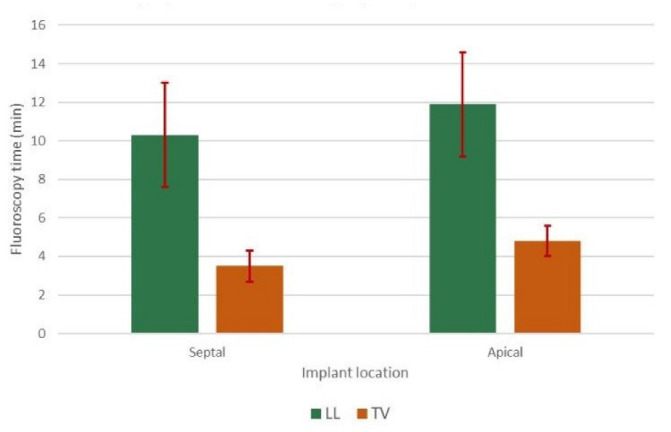
Fluoroscopy times in septal vs. apical implants for LL and TV PMs.

**Figure 2 jcm-11-06071-f002:**
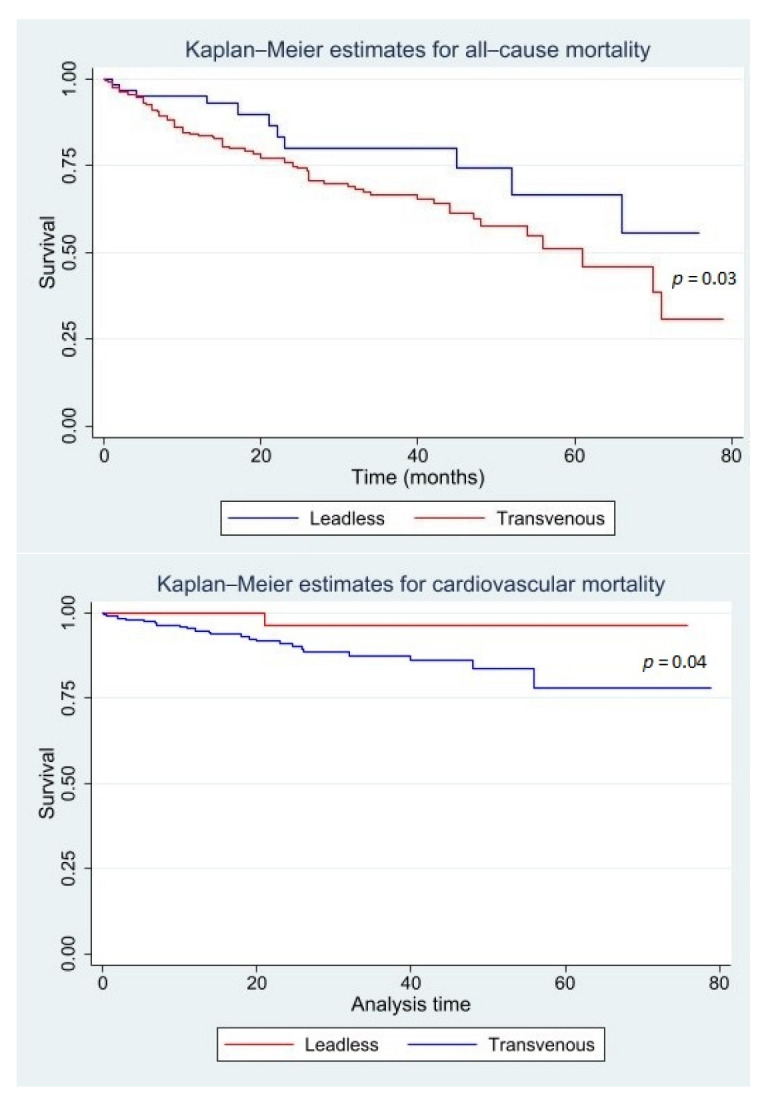
Kaplan–Meier estimates of all-cause and cardiovascular mortality in LL and TV PM recipients over the follow-up period.

**Figure 3 jcm-11-06071-f003:**
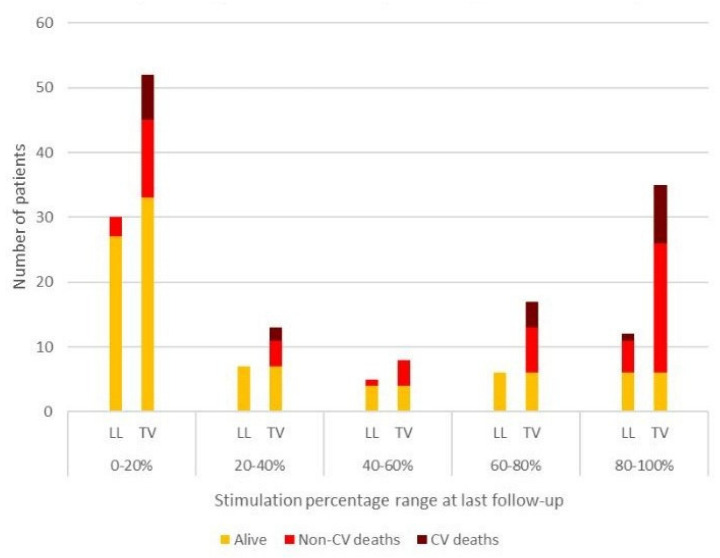
Cardiovascular and non-cardiovascular mortality in LL and TV PM recipients weighed against stimulation percentage at last follow-up.

**Table 1 jcm-11-06071-t001:** Baseline patient characteristics and 30-day perioperative complications in LL and TV PM recipients.

	Leadless (n = 72)	Transvenous (n = 272)	Significance (*p* < 0.05)
**Baseline characteristics**			
Age; median [SEM *]	79.5 [2.5]	85.0 [1.0]	***p* < 0.01**
Sex (female)	26/72 (36%)	111/272 (41%)	*p* = 0.47
Diabetes mellitus	19/72 (26.4%)	49/272 (18.0%)	***p* < 0.01**
Hypertension	51/72 (70.8%)	212/272 (77.9%)	*p* = 0.21
Ejection fraction; median [SEM]	57 [3]%	59 [2]%	*p* = 0.49
Permanent atrial fibrillation	58/72 (80.6%)	262/272 (96.3%)	***p* < 0.01**
Ischaemic heart disease	14/72 (19.4%)	83/272 (30.5%)	*p* = 0.06
Previous CIED extraction	7/72 (9.7%)	2/272 (0.7%)	***p* < 0.01**
Surgical or percutaneous treatment of valvular disease	20/72 (27.8%)	82/272 (30.1%)	*p* = 0.70
Chronic kidney disease ** (GFR < 60 mL/min/1.73 m^2^)	26/72 (36.1%)	157/272 (57.7%)	***p* < 0.01**
Chronic haemodialysis	5/72 (6.9%)	2/272 (0.7%)	***p* < 0.01**
Bedridden/cognitive impairment	0/72 (0%)	3/272 (1.1%)	*p* = 0.37
**30-day perioperative complications**			
Pericardial effusion	0/72 (0%)	4/272 (1.5%)	*p* = 0.30
Tamponade	0/72 (0%)	1/272 (0.4%)	*p* = 0.61
Lead/device dislocation	1/72 (1.4%)	0/272 (0%)	*p* = 0.06
Pneumothorax [requiring drainage]	/	6/272 (2.2%) [1/272 (0.4%)]	/
Haematoma [requiring surgical revision]	3/72 (4.2%) [0/72 (0%)]	3/272 (1.1%) [1/272 (0.4%)]	*p* = 0.08
Overall	4/72 (5.6%)	14/272 (5.1%)	*p* = 0.33

* SEM: standard error of the median; ** eGFR < 60 mL/min/1.73 m^2^.

**Table 2 jcm-11-06071-t002:** Electrical parameters after implantation and patient data/electrical parameters at follow-up.

	Leadless (n = 72)	Transvenous (n = 272)	Significance (*p* < 0.05)
**Electrical parameters after implantation**			
Mean capture threshold (V × 0.4 ms)	0.75 ± 0.08	0.69 ± 0.04	*p* = 0.79
Mean impedance (Ohm)	748 ± 28	698 ± 15	*p* = 0.06
Mean intrinsic R wave amplitude (mV)	9.8 ± 0.6	10.8 ± 0.4	*p* = 0.58
**Electrical parameters at follow-up**			
Mean capture threshold (V × 0.4 ms)	0.62 ± 0.04	0.79 ± 0.03	***p* = 0.005**
Mean capture threshold increase > 1 V	2/72 (2.7%)	9/262 (3.4%)	*p* = 0.12
Mean impedance (Ohm)	636 ± 18	606 ± 14	***p* = 0.009**
Mean intrinsic R wave amplitude (mV)	11.6 ± 0.5	12.0 ± 0.4	*p* = 0.86
**Patient data at follow-up**			
Mean follow-up time (months)	22.8 ± 2.6	23.7 ± 1.1	*p* = 0.31
Superficial Suture Infection [requiring surgical revision]	0/72 (0%)	3/272 (1.1%) [2/272 (0.8%)]	*p* = 0.37
Haematoma	0/72 (0%)	1/272 (0.4%)	*p* = 0.61
Skin sore	0/72 (0%)	1/272 (0.4%)	*p* = 0.61
Overall long-term complications	0/72 (0%)	5/272 (1.9%)	*p* = 0.25
Pacing system revisions [Repositioning] [Lead addition]	0/72 (0%)	6/272 (2.3%) [5/272 (1.9%)] [1/272 (0.4%)]	*p* = 0.20

**Table 3 jcm-11-06071-t003:** Multivariate analysis of all-cause mortality in LL and TV cohorts and on the entire population.

**Multivariate Analysis of All-Cause Mortality in LL vs. TV**				
**Variable**	**Leadless** **(n = 72)**		**Transvenous** **(n = 272)**	
	**Odds ratio**	***p* value**	**Odds ratio**	***p* value**
Age	1.01 [0.92–1.11]	0.78	0.92 [0.87–0.97]	**<0.01**
Female sex	0.76 [0.11–5.05]	0.77	0.52 [0.26–1.03]	0.06
Diabetes mellitus	1.13 [0.13–9.72]	0.91	0.45 [0.20–1.02]	0.05
Chronic kidney disease *	0.74 [0.11–5.07]	0.76	0.71 [0.33–1.48]	0.36
Ischaemic heart disease	0.45 [0.08–2.63]	0.38	1.16 [0.57–2.35]	0.69
Left ventricular ejection fraction	1.02 [0.93–1.12]	0.68	1.01 [0.98–1.05]	0.42
Ventricular stimulation percentage	0.26 [0.06–1.11]	0.07	1.00 [0.99–1.01]	0.73
**Multivariate analysis of all-cause mortality in both cohorts**				
**Variable**	**Odds ratio**		***p* value**	
Leadless vs. transvenous	1.42 [0.56–3.56]		0.46	
Age	0.94 [0.90–0.98]		**0.01**	
Female sex	0.57 [0.31–1.06]		0.08	
Diabetes mellitus	0.57 [0.28–1.16]		0.12	
Chronic kidney disease *	0.62 [0.32–1.20]		0.16	
Ischaemic heart disease	0.97 [0.52–1.82]		0.93	
Left ventricular ejection fraction	1.01 [0.98–1.04]		0.58	
Percentage ventricular stimulation	1.00 [0.99–1.01]		0.82	

* eGFR < 60 mL/min/1.73 m^2^.

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
