# Peer review of "Is Less Always More? A Prospective Two-Centre Study Addressing Clinical Outcomes in Leadless versus Transvenous Single-Chamber Pacemaker Recipients"

_jcm, 2022, doi:10.3390/jcm11206071_

Round 1

Reviewer 1 Report

Although the topic of the article is not quite my field of activity, I think that the ideea is an interesting one, which, although it is not necessarily something new, practically highlights the experience of the authors in the field of permanent electrical cardiostimulation.

Although the group of patients is not very large, the experience of the authors can be taken into account and can be the basis of wider research on the same topic. Many ussefull practical and technical details are provided.

 Some of my comments are inserted directly into the text

Author Response

Many thanks for your prompt feedback on our work.  I shall address in order the points raised in your review: 

  • "in sinus rhythm": in the context of complete atrioventricular block VVIR indication included either patients in atrial fibrillation or patients in sinus rhythm (with complete heart block) who were bedridden/cognitively impaired (where the addition of an additional atrial lead had an unfavourable risk/balance
  • "pace-maker syndrome and heart failure": absolutely interesting point to discuss. Unfortunately, echocardiography data at follow-up was not available consistently across cohorts thus complicating any conclusions on the matter. Given the differences in age and comorbidities between the two cohorts reflected in different pacing percentages (Figure 3), we would probably expect a higher incidence of pacemaker syndrome in the TV group relating however to the differences in pacing requirement rather than differences between TV and LL per se. With regards to heart failure, unfortunately no consistent data on HF events was collected at follow-up. Our impression is however that these were rather limited in both cohorts given the relatively minor contribution of cardiovascular causes to mortality in both cohorts.

I have proceeded to edit the manuscript as you suggested (you can find a copy of the edited manuscript attached). 

Reviewer 2 Report

This study included 344 patients who received implantation of a VVIR pacemaker. The aim of this study is to prospectively analyze survival and complication rates in leadless versus transvenous VVIR pacemakers. The investigators showed that the complication rates between two groups were not significantly different. The transvenous recipients had greater all-cause and cardiovascular mortality. The investigators concluded that the low complication rates and short life expectancy of both groups provide more evidence against extending leadless pacemaker indications to all VVIR candidates. The paper is well written, and the key point is clinical relevant.

There are still some concerns:

1.       The presentation of continuous values is not confusing, e. g. age 79.5 ± 2.5 is the presentation of average rather than median. The presentations and comparisons of data are not completely compatible.

2.       Although the LL group had better survival outcomes than the TV group, the comparisons are not justified because the baseline parameters between two groups are significantly different. Do you think that the propensity-score matching analysis works?

3.       In conclusions: “Based on our patient selection criteria, these may account for 290 approximately 20 percent of VVIR recipients.” I am curious how did you find the percentage of good candidate according to the selection criteria? Would you like to mention that in the Methods and Results?

Author Response

Many thanks for your review. I shall address the points raised in order: 

1) Wherever median values (rather than mean) are presented I have removed the +/- symbol and specificied that the standard error of the median is being used.

2) Propensity score matching: this is indeed a possibility we did consider when analysing the data. However, given the rather limited impact played by individual comorbidities in influencing mortality (as demonstrated by the multivariate analysis) with only age significantly impacting survival, we believed this analysis would not add any further useful information. In addition, we decided from the outset that the patient profile for LL and TV would be different given the individualised device selection criteria set out for LL vs. TV VVIR PMs (as explained in the Materials and Methods). Thus, a propensity score matching, aiming to mimick a randomisation process, would have been conceptually against our study design. Furthermore, the significant difference in sample size (with far less LL patients than TV) would have complicated further this type of analysis. 

3) 20% figure: this figure was not actually set out when deciding on LL vs. TV indication but rather we observed that, after using the individualised patient selection (detailed in the Materials and Methods section), the percentage of patients with LL indication was roughly 20%. When specifying this in our conclusions, we do not attempt to set this as a "hard rule" but rather to suggest that in the two tertiary cardiology centres involved in this study when using the above LL selection criteria (aiming to minimise overall VVIR PM complications), roughly 20% of patient have an indication for LL pacing. This proportion may of cours differ in other centres with different patient demographics.

Round 2

Reviewer 2 Report

About the “20% of VVIR recipients”, my suggestion is NOT to specify the data in the Conclusions since you did not really analyze the data in this study.

Author Response

Many thanks for your review. I have proceeded to edit the manuscript as you suggested removing the "20% of VVIR recipients" comment from both the abstract and the conclusions. 

Thank you for your feedback.
